# Positive Impacts of Aphanizomenon Flos Aquae Extract on Obesity-Related Dysmetabolism in Mice with Diet-Induced Obesity

**DOI:** 10.3390/cells12232706

**Published:** 2023-11-25

**Authors:** Simona Terzo, Pasquale Calvi, Marta Giardina, Giacoma Gallizzi, Marta Di Carlo, Domenico Nuzzo, Pasquale Picone, Roberto Puleio, Flavia Mulè, Stefano Scoglio, Antonella Amato

**Affiliations:** 1Department of Biological-Chemical-Pharmaceutical Science and Technology, University of Palermo, 90128 Palermo, Italy; simona.terzo01@unipa.it (S.T.); pasquale.calvi@unipa.it (P.C.); marta.giardina@community.unipa.it (M.G.); flavia.mule@unipa.it (F.M.); 2Department of Biomedicine, Neuroscience and Advanced Diagnostic, University of Palermo, 90127 Palermo, Italy; 3Istituto per la Ricerca e l’Innovazione Biomedica (IRIB), CNR, Via U. La Malfa 153, 90146 Palermo, Italy; giacoma.galizzi@irib.cnr.it (G.G.); marta.dicarlo@irib.cnr.it (M.D.C.); domenico.nuzzo@irib.cnr.it (D.N.); pasquale.picone@irib.cnr.it (P.P.); 4Istituto Zooprofilattico Sperimentale della Sicilia “A. Mirri”, Via Gino Marinuzzi 3, 90129 Palermo, Italy; roberto.puleio@izssicilia.it; 5Centro di Ricerche Nutriterapiche, 61029 Urbino, Italy; stefanoscoglio@me.com

**Keywords:** Klamath algae, high fat diet, insulin resistance, adiposity, inflammation, oxidative stress

## Abstract

The present study evaluated the ability of KlamExtra^®^, an Aphanizomenon flos aquae (AFA) extract, to counteract metabolic dysfunctions due to a high fat diet (HFD) or to accelerate their reversion induced by switching an HFD to a normocaloric diet in mice with diet-induced obesity. A group of HFD mice was fed with an HFD supplemented with AFA (HFD-AFA) and another one was fed with regular chow (standard diet—STD) alone or supplemented with AFA (STD-AFA). AFA was able to significantly reduce body weight, hypertriglyceridemia, liver fat accumulation and adipocyte size in HFD mice. AFA also reduced hyperglycaemia, insulinaemia, HOMA-IR and ameliorated the glucose tolerance and the insulin response of obese mice. Furthermore, in obese mice AFA normalised the gene and the protein expression of factors involved in lipid metabolism (FAS, PPAR-γ, SREBP-1c and FAT-P mRNA), inflammation (TNF-α and IL-6 mRNA, NFkB and IL-10 proteins) and oxidative stress (ROS levels and SOD activity). Interestingly, AFA accelerated the STD-induced reversion of glucose dysmetabolism, hepatic and VAT inflammation and oxidative stress. In conclusion, AFA supplementation prevents HFD-induced dysmetabolism and accelerates the STD-dependent recovery of glucose dysmetabolism by positively modulating oxidative stress, inflammation and the expression of the genes linked to lipid metabolism.

## 1. Introduction

The rapid epidemiological and nutritional transitions occurring in much of the world over the last 50 years has led to a global obesity epidemic, contributing to the progressive increase in the prevalence of diabetes and other diet-related chronic diseases such as cardiovascular diseases, fatty liver disease and cancer [1].

The constant positive energy balance contributes to the development of obesity, as excess energy is efficiently deposited in the form of triglycerides in adipose tissue through the lipogenic pathway. The dramatic increase in fat mass is responsible for hyperplasia and hypertrophy development in the visceral adipose tissue. Hypertrophic adipocytes have been shown to be associated with an increased rate of lipolysis, pro-inflammatory cytokines release and reactive oxygen species (ROS) production, that, in turn, are involved in insulin resistance (IR) and non-alcoholic fatty liver disease (NAFLD) development [2]. The pro-inflammatory cytokine hypersecretion participates in IR development by attenuating insulin receptor substrate 1 (IRS-1)-mediated insulin signalling [3], and it also promotes adipocyte lipolysis leading to augmented levels of serum free fatty acids (FFA) in the bloodstream. FFA and glycerol hydrolysed from visceral adipocytes are transported to the liver, contributing to hepatic fat accumulation and steatosis development. Excessive hepatic lipid accumulation in obesity can also be caused by other different metabolic perturbations such as excessive dietary FFA, new hepatic FFA synthesis through de novo lipogenesis or reduced export of Very Low-Density Lipoproteins (VLDL). The increased hepatic lipid deposition induces high rates of β-oxidation, increased ROS production, activation of pro-inflammatory pathways, oxidative stress and cellular damage [4].

Among the common approaches for the weight control are lifestyle interventions and the pharmacotherapy, but their effectiveness is usually compromised because of poor compliance. In fact, a long-term caloric restriction could compromise diet adherence and reduce weight loss success and the related improvement of metabolic dysfunctions [5]. In recent years, scientific research has been heavily dedicated to the identification of foods or natural supplements with better-tolerated anti-obesity properties [6,7,8]. An alternative font of functional foods and nutraceuticals is represented by microalgae which are a good option for developing foods that are useful in metabolic disorders management [9,10,11]. Microalgae contain minerals, vitamins and metabolites with nutritional antioxidant, anti-inflammatory and neuromdulating properties, offering a large spectrum of possible applications and utilizations [12]. The market for microalgae nutraceuticals is dominated by two cyanobacteria, *Spirulina* and *Aphanizomenon flos-aquae* (AFA). AFA grows in the Upper Klamath Lake in Oregon, and it is known as Klamath algae. While different types of experimental data clearly show health positive effects of *Spirulina* [13], the functional effects of Klamath algae have not been fully evaluated. Klamath is rich in natural antioxidants, organic minerals, vitamins, polyunsaturated fatty acids (PUFA), amino acids and enzymes [14]. Klamath algae also contains phenylethylamine (β-PEA), an important neuromodulator, by which Klamin^®^, an AFA extract that concentrates β-PEA, exerts protective effects against neurological diseases [15,16], neuroinflammation [14] and intestinal dysmotility [17]. Klamath algae is also source of a unique type of phycocyanins (AFA-phycocyanins) significantly concentrated in another AFA extract, named AphaMax^®^. Phycocyanins are known for the antioxidant, anti-inflammatory [18,19], and antiproliferative [20] properties. AFA algae is also a significant source of chlorophyll and omega 3 fatty acids, which exert significant anti-inflammatory activity by suppressing pro-inflammatory cytokines production [21,22,23]. Thus, the direct application of Klamath algae as a nutritional supplement might be a promising strategy in counteracting obesity-related metabolic disorders. KlamExtra^®^ is a new combination of two patented AFA extracts: Klamin^®^, showing neuro- and immunomodulatory properties [14,15,16], and AphaMax^®^, having antioxidant and anti-inflammatory properties [18,19,20]. KlamExtra^®^ supplementation is addressed to support the homeostasis of the cardiovascular, nervous and immune systems [24]. Recent data revealed that KlamExtra^®^ is able to counteract the neuronal damages caused by HFD through decreasing neuroinflammation [25]. In particular, in the present study we evaluated the ability of KlamExtra^®^ to ameliorate obesity-related metabolic dysfunctions induced by chronic exposure to an hyperlipidic diet and to potentiate the reversion of HFD-induced dysmetabolism obtained by switching an HFD to a normocaloric diet (standard diet-STD) in a mouse model of diet-induced obesity. The KlamExtra^®^ effects were evaluated against glucose and lipid dysmetabolism, hepatic steatosis and adiposity, liver and adipose tissue inflammation, oxidative stress and abnormal lipid metabolizing gene expression induced by HFD. From now on, KlamExtra^®^ will be indicated as AFA.

## 2. Materials and Methods

### 2.1. Animals and Experimental Protocols

All procedures used complied with the Italian legislative decree N° 26/2014, and the European directive 2010/63/UE. The experimental protocols were approved and authorised by the Italian Ministry of Health (Rome, Italy; Authorization Number 46/2020-PR).

Forty male C57BL6/J (Envigo Laboratories, San Pietro Al Natisone Udine, Italy) mice were purchased at 4 weeks of age and were acclimated to a 12:12 h light–dark cycle for 1 week, consuming regular chow and water freely. After acclimation, mice were randomly divided into different groups as described in Figure 1: Lean control (Lean; *n* = 8), fed a regular standard diet (code 4RF25, Mucedola, Milan, Italy) for 18 weeks, and HFD obese control (HFD; *n* = 32), fed a high fat diet with 60% of caloric intake deriving from fat (HFD; code PF4051/D, Mucedola) for 10 weeks to establish obesity and metabolic dysfunctions [26]. At the end of the 10 weeks, after having ascertained the development of the HFD-dependent dysmetabolic state, obese mice were further subdivided into four groups: one group fed an HFD (HFD) for a further 8 weeks; another one fed with an HFD supplemented with AFA (HFD-AFA; *n* = 8) for 8 weeks; and another group had their diet switched from an HFD to a standard diet, alone (STD; *n* = 8) or supplemented with AFA (STD-AFA; *n* = 8), for further 8 weeks. 

The HFD and STD diets supplemented with AFA were custom designed and prepared by Mucedola S.R.L, by adding 8,33 g/Kg of KlamExtra^®^ for both HFD-AFA (code PF20432; Mucedola S.R.L, Milan, Italy) and STD-AFA (code PF20458; Mucedola S.R.L, Milan, Italy). AFA extract composition is reported in Appendix A. 

The KlamExtra^®^ dose was chosen based on the humane dosage, and it corresponds to 25 mg AFA ingested/day/mouse. Food intake and body weight were recorded at each 1-week interval. The percentage of weight changes was calculated following the formula: starting weight from the switch diets minus weekly weight/(starting weight) × 100. Food intake was calculated according to the weight change of the chow box in an interval of 24 h. 

At the end of the 23 weeks, in vivo analyses were performed, and then all animals were weighed and sacrificed by cervical dislocation, and the liver and visceral adipose tissue (VAT) were removed for necropsy. Blood was collected via cardiac puncture and transferred into tubes containing ethylenediaminetetraacetic acid (EDTA) 1 mg/mL. Samples were centrifuged at 3000 rpm for 10 min and the obtained plasma stored at −80 °C. VAT and liver were weighed, and one part of each tissue was stored at −80 °C until biomolecular analysis; the second part was collected in 4% neutral formalin solution for histological analysis.

### 2.2. Biochemical Analyses

The blood cholesterol and triglyceride concentrations were determined using the analyser MultiCare (Biochemical Systems International-Srl, Arezzo, Italy). The glucose levels were measured in a drop of blood from the tail vein with a glucometer (GlucoMen LX meter, Menarini, Italy). Levels of insulin in the plasma were measured using an ELISA kit (Alpco diagnostics, Salem, NH, USA). We evaluated both glucose and insulin sensitivity with the Intraperitoneal glucose tolerance test (IPGTT) and insulin tolerance test (ITT), respectively. Mice fasting for 16 h received an intraperitoneal (i.p.) injection of glucose (2 g/kg body weight) in 0.9% saline for IPGTT, or insulin (0.5 U/kg body weight) (Lantus 100 UI/10 mL, code 03572421; Sanofi, Milano, Italy) in 0.9% saline for ITT. Then, blood glucose was measured at different time intervals (0, 15, 30, 60, and 120 min from i.p. injection). The Homeostasis Model Assessment of basal Insulin Resistance (HOMA-IR), an index of insulin resistance, was obtained as described below: fasting glucose (mg/dL) × fasting insulin (mU/L) divided by the constant 22.5.

The liver lipids were extracted from the fresh liver homogenate using Folch’s method as previously described [27], and the extracts were evaporated in a vacuum and weighed. 

### 2.3. Liver and Adipose Tissue Histology

Liver and VAT were fixed in 4% formaldehyde, washed thoroughly, subsequently dehydrated in ethanol and embedded in paraffin. Consecutive sections (5 µm) were mounted on slides and stained with haematoxylin/eosin. Under the light microscope, 5 randomly chosen liver and adipose tissue fields were analysed. The section images were acquired with a light microscope (Leica DMLB, Meyer instruments, Houston, TX, USA) provided with a DS-Fi1 camera (Nikon, Florence, Italy) and 10× and 20× magnification were used. The staging of liver steatosis was based on the amount and types of the fat (macrovesicular and microvesicular), and defined as absent, light, moderate or severe when 1%, 30%, 30–60% or 60% of the hepatocytes were, respectively, involved. Adipose cell size was calculated with image analysis software Basic research NIS elements F 2.30 (Nikon, Florence, Italy), by manual tracing of at least 500 adipocytes for each field.

### 2.4. Real-Time PCR

Total RNA from liver and VAT was extracted using the High Pure RNA Isolation Kit (Roche, Vienna, Austria) following the manufacturer’s instructions. The mRNA was reverse transcribed using a QuantiNova Reverse Transcription kit (Qiagen, Milan, Italy). Real-time PCR reactions were then performed using the PowerTrack SYBER Green Master Mix (Applied Biosystem, Foster City, CA, USA) on a StepOne Real-Time (Applied Biosystem, Foster City, CA, USA), in triplicate. The following thermal cycling conditions were used: 2 min of initial denaturation at 95 °C, followed by 45 cycles of one-step PCR denaturation at 95 °C for 15 s and annealing/extension at 60 °C for 30 seconds. The specificity of the real-time PCR was verified via melting-curve analysis of each PCR reaction. Primers specific to mouse GAPDH were used as the control to normalise loading. The levels of the target mRNAs were determined using the 2^−ΔΔCt^ values between the target and loading control. The primers used for real-time PCR are listed in the Table 1.

### 2.5. Semi-Quantitative RT-PCR

Total RNA from liver and VAT was used also for semi-quantitative RT-PCR. For RT-PCR amplification a Taq polymerase mix was used (yourSIAL Taq HS Mix, Sial, Rome, Italy) in which 10 µM forward and reverse primers specific for the fatty acid synthase (FAS), peroxisome proliferator-activated receptor gamma (PPAR-γ), Sterol Regulatory Element-Binding Protein (SREBP)-1c, Fatty acid transporter (FAT)-P and β-actin genes, were added, to a final volume of 25 µL. The primer sequences of the genes investigated are presented in Table 2. The PCR products’ size was visualised using an ultra-violet lamp following electrophoresis in 1.8% agarose gel stained with ethidium bromide. Intensities of the PCR bands were analysed densitometrically using E-Gel GelQuant Express Analysis Software (version 1.14.6.0 Dongle-Thermo Fisher Scientific, Monza, Italy).

### 2.6. Western Blot Analysis

Liver and VAT samples were homogenized on the ice-cold buffer (50 mM Tris-HCl pH 7.4, 0.5% Triton X-100, 150 mM NaCl, 1 mM DTT, 2 mM PMSF, 0.1% SDS) with protease and phosphatase inhibitor cocktail II (Amersham, Life Science, Les Ulis, France and Sigma-Aldrich, Poole, Dorset, UK, respectively). The Bradford method (Bio-Rad, Segrate, Italy) was used to quantify total proteins. Then, 50 μg of proteins were separated with 10% or 12% acrylamide gel and then transferred onto nitrocellulose filter. The filter was incubated with anti-interleukin 10 (IL-10, 1:1000, Santa Cruz Biotechnology, Santa Cruz, CA, USA), anti-NFkB p65 (1:1000, Santa Cruz Biotechnology, Santa Cruz, CA, USA), Insulin-Receptor (1:1000, Thermo Fisher Scientific, MA USA) and anti-β-actin (1:10,000, Sigma-Aldrich, St. Louis, MO, USA). The Odyssey^®^ scanner (LI-COR Biosciences, Lincoln, NE, USA) was used to detect primary antibodies, according to the manufacturer’s instructions, using anti-mouse and anti-rabbit secondary antibodies labelled with IR790 and IR680 (1:10,000; Life Technology, Carlsbad, CA, USA). Band intensities were analysed with ImageJ software 1.53 m and expression was normalised to the β-actin expression. 

### 2.7. Detection of Oxidative Levels: DCFH-DA Assay 

By using dichlorofluorescein diacetate, (DCFH-DA; Sigma-Aldrich, St. Luois, MO, USA) the ROS levels in livers and VAT were measured, according to the method described by Mudò [28], with some modifications. Briefly, 10 mg of liver or VAT was suspended on 1000 μL of PBS1× with 10 μL of protease inhibitors (Amersham Life Science, Les Ulis, France). Samples were added to the dichlorofluorescein diacetate (DCFH-DA-1 mM) for 10 min, at room temperature and in the dark. Then, the samples were washed with PBS, and analysed with a fluorimeter (Glomax Microplate reader; Promega Italia Srl, Milano, Italy) by setting the excitation filter at 485 nm and the emission filter at 530 nm. Relative ROS generation was expressed as the change in fluorescence among the experimental groups compared to the respective control group (100%).

### 2.8. SOD Activity

Liver and VAT were separated in PBS with protease inhibitors (Amersham Life Science; Milano, Italy). The homogenized tissues were sonicated and centrifuged (14,000 rpm, at 4 °C, for 30 min) to remove insoluble material. Total proteins obtained from supernatant were quantified using the Bradford method. According to manufacturer’s instructions, 50 μg of protein was used for total SOD enzymatic activity quantification, and the SOD assay kit (Sigma-Aldrich, St. Luois, MO, USA). Glomax Microplate Reader (Promega Italia Srl, Milano, Italy) was used to read absorbance by setting the filter to 450 nm.

### 2.9. Statistical Analysis

Statistical analysis was performed using Prism 6.0, GraphPad (San Diego, CA, USA). Comparisons between the three experimental groups were made by using ANOVA followed by the Bonferroni post hoc test. Differences were considered statistically significant when p < 0.05. Results were expressed as mean ± SEM.

## 3. Results

### 3.1. Effects of 10 Weeks of HFD on Metabolic Parameters

After ten weeks, mice fed with the HFD displayed higher body weights, calorie intakes and blood lipid levels than the lean mice group (Figure 2A–D). HFD mice also showed hyperglycaemia and altered glucose and insulin sensitivity compared to the lean mice (Figure 2E–I), confirming that ten weeks of a hyperlipidemic diet are enough to affect metabolic homeostasis. Then, HFD mice were sub-divided in four groups and were differently fed, with or without AFA extract, for a further 8 weeks.

### 3.2. Impact of AFA on Body Weight, Food Intake and Lipidaemia

The hyperlipidic diet for 18 weeks exacerbated obesity, increased caloric intake and dyslipidemia in HFD animals compared to the lean mice (Figure 3A–D). Although AFA supplementation started at the 10th week of HFD, the final body weight, the weight gain and the triglyceride levels of HFD-AFA mice were significantly reduced compared to the HFD animals (Figure 3A,B). On the contrary, AFA did not potentiate the reductions in body weight, weigh loss or blood levels of lipids observed in STD animals (Figure 3A,B,D). Moreover, AFA diets did not affect food intake of either HFD or STD mice (Figure 3C). 

### 3.3. Impact of AFA on Glucose Homeostasis

AFA supplementation exerted positive effects on glucose metabolism. Indeed, HFD-AFA mice had a significant reduction in their fasting glucose concentration (Figure 4A) and an improvement in their glycemic response, as shown by the lower blood glucose levels during the GTT (Figure 4B,C) in comparison with obese animals. AFA supplementation also prevented HFD-dependent hyperinsulinemia (Figure 4D). Accordingly, the insulin sensitivity test demonstrated that the reduction in insulin sensitivity observed in HFD mice was significantly ameliorated in the HFD + AFA group (Figure 4E,F). Moreover, the HOMA-IR was lower in HFD + AFA animals compared to HFD, becoming similar to lean mice (Figure 4G). These results reveal that AFA extract is potentially able to counteract the HFD-induced insulin-resistance. Western blot analysis strengthened this hypothesis, revealing a reduced expression of the insulin receptor in the HFD-AFA livers compared to HFD (Figure 4H,I). 

AFA supplementation also potentiated an improvement in glucose metabolism induced by STD consumption. In fact, STD-AFA mice showed fasting glucose levels (Figure 4A) and area under curve (AUC) for ITT, HOMA-IR and hepatic IR protein expression that were lower than the STD group and similar to that of the lean mice (Figure 4E–I). Although AFA supplementation did not further ameliorate the glucose tolerance (Figure 4B,C). 

### 3.4. Impact of AFA on Hepatic Steatosis and Adiposity

Liver histological analysis (Figure 5A) showed normal liver parenchyma with absence of steatosis in lean mice. Conversely, the HFD livers showed severe steatosis with intra- and extrahepatic fat vacuoles accumulation. Interestingly, AFA supplementation was able to prevent the HFD-induced severe grading of steatosis. In fact, in the HFD-AFA livers, lipid droplets were partially suppressed, showing a steatosis grading as moderate. Accordingly, the absolute and relative (%) liver weight and the intrahepatic lipid content were markedly reduced in the animals fed an HFD supplemented with AFA compared to obese controls (Figure 5B–D).

AFA was not able to induce further improvements in the steatosis reversion induced by the standard diet. In fact, both livers of STD and STD-AFA mice showed an absence of steatosis (Figure 5A) and the same degree of reduction for intrahepatic lipid content and the absolute and relative liver weight ratio (Figure 5B–D).

VAT histological analysis showed that the adipocytes area in HFD mice was significantly higher than that in lean mice, suggesting hypertrophy (Figure 6A,B). Adipose tissue weight and the fat index were increased in HFD mice when compared to the lean group (Figure 6B–D). However, these increments were significantly reduced by the HFD-AFA diet, suggesting that the AFA supplementation counteracts HFD-induced hypertrophy. Moreover, the adipocyte size distribution analysis revealed that, in HFD-AFA mice, the major cell distribution was for adipocytes with a smaller area (9000–10,000 µm^2^) with respect to HFD adipocytes, in which the major cell distribution was achieved for adipocytes with an area of >15,000 µm^2^ (Figure 6E).

The adipocyte size, adipose tissue weight, and fat index reduction observed in STD mice was similar to that observed in the adipose tissue of the STD-AFA group, and similar to the lean group, suggesting that AFA supplementation did not potentiate STD-induced hypertrophy regression. 

### 3.5. Impact of AFA on the Lipid Metabolizing Gene Expression in the Liver and Adipose Tissue

Our results clearly revealed a hypolipidemic effect of AFA in the liver and adipose tissue of obese mice. So, we evaluated the impact of AFA supplementation on the lipid metabolism-related gene expression changes in liver and adipose tissue of lean, HFD and HFD-AFA, by using RT-PCR analysis.

As shown in Figure 7, FAS, PPAR-γ, SREBP-1c and FAT-P mRNA levels were overexpressed in the liver and VAT of obese mice in comparison with the lean group (Figure 7A–D). In the livers of HFD-AFA mice a slight reduction in PPAR-γ and FAT-P, and a significant decrease in SREBP-1c mRNA levels compared to HFD, were observed (Figure 7A,B). In the HFD-AFA adipose tissue, FAS, SREBP-1c and FAT-P mRNA levels were strongly reduced in comparison with the obese controls (Figure 7C,D).

### 3.6. Impact of AFA on Hepatic and Adipose Tissue Inflammation

In order to clarify if AFA counteracted glucose dysmetabolism by ameliorating inflammatory status, the gene and protein expression of several inflammatory markers were investigated in the livers and VAT of the different groups of animals. Gene expression analysis revealed an up-regulation of TNF-α and IL-6 in both the livers and adipose tissues of obese animals compared to lean controls. At the same time, the levels of the proteins NFkB and IL-10 resulted in dysregulated HFD livers and adipose tissues compared to the lean group (Figure 8A–F). All of these results confirmed the presence of inflammation in the HFD tissues. AFA supplementation significantly ameliorated hepatic inflammation. In fact, the HFD-AFA livers showed TNF-α and IL-6 gene expression levels that were significantly downregulated compared to HFD mice, and NFkB and IL-10 protein levels significantly improved in comparison with HFD livers, similar to that observed in the lean controls. A slight but not significant improvement in the TNF-α and IL-6 gene expression and the protein levels of NFkB and IL-10 were also observed in the HFD-AFA adipose tissue compared to the HFD mice (Figure 8A–F). Interestingly, AFA supplementation was also able to potentiate the STD-induced regression of inflammation; in fact, a significant reduction in IL-6 and TNF-α gene expression was observed in the livers and adipose tissues of STD-AFA mice in comparison with STD mice (Figure 8A,D). No improvement was observed in the hepatic and VAT NFkB and IL-10 protein expression between the STD-AFA and STD groups (Figure 8E,F). 

### 3.7. Impact of AFA on Oxidative Stress in Liver and Adipose Tissue

The chronic consumption of a hyperlipidic diet is related to increased oxidative stress. Accordingly, the liver and the adipose tissue from obese animals showed increased levels of ROS (Figure 9A–C) and reduced SOD activity (Figure 9B–D) in comparison with lean mice. Both AFA diets showed positive effects against oxidative stress. In fact, ROS levels were significantly reduced in the livers and adipose tissues of the HFD-AFA mice compared to obese mice, and in STD-AFA mice in comparison with STD animals (Figure 9A–C). Hepatic SOD activity was significantly ameliorated in HFD-AFA mice compared to obese controls, while no further improvement for the enzymatic activity was observed between the livers of STD-AFA and STD mice (Figure 9B). SOD activity of adipose tissue was not modified by the different diets.

## 4. Discussion

The results of the present study show that the supplementation with the Aphanizomenon flos aquae (AFA) KlamExtra^®^ is able to alleviate obesity-related dysmetabolism and to potentiate the improvement of the HFD-induced metabolic abnormalities obtained by switching from hyperlipidic diet to a low-fat diet in mice with diet-induced obesity. AFA exerts its beneficial effects by positively modulating the expression of factors linked to inflammation, oxidative stress and lipid metabolism. 

During the last few decades, developed and developing countries have experienced a significant shift in food habits, characterised by the high consumption of high fat, hypercaloric and salty foods [29,30]. This unhealthy nutrition predisposes people to various chronic non-communicable diseases, including obesity and related metabolic dysfunctions such as hyperglycaemia, impaired glucose response, insulin resistance, hepatic steatosis and adiposity [31,32,33]. The most common therapeutic approach against obesity and the related disorders involves lifestyle interventions, such as hypocaloric diets and increased physical activity. The effectiveness of hypocaloric diets is often linked to the consumption of foods or nutraceuticals rich in antioxidant and anti-inflammatory phytochemicals [34,35,36]. 

An interesting source of bioactive anti-obesity compounds is represented by microalgae. Recently, we demonstrated that an Aphanizomenon flos aquae (AFA) extract, KlamExtra^®^, is able to counteract neurodegeneration in obese mice [25]. Here, we investigated the impact of KlamExtra^®^ supplementation on obesity-related metabolic dysfunctions in mice with HFD-induced obesity. Mice fed with a HFD develop obesity and a plethora of metabolic dysfunctions similar to human metabolic syndrome such as weight gain, glucose and lipid impaired metabolisms, hepatic steatosis, adipose tissue hypertrophy, increased inflammation and oxidative stress [37,38]. To analyse the efficacy of AFA to counteract the progression of the dysmetabolism induced by a chronic hyperlipidic diet, or to accelerate healing from obesity due to hypocaloric diet, we started with 8 weeks of AFA supplementation in mice with obesity induced by 10 weeks of HFD feeding. The achievement of the dysmetabolic condition was confirmed by an in vivo analysis performed in HFD mice at the end of the 10 weeks of hyperlipidic diet. 

### 4.1. Does AFA Prevent the Long-Term HFD Dependent Metabolic Dysfunctions?

First, we demonstrated that AFA supplementation prevents HFD-induced weight gain, as suggested by the lower final body weight and reduced weight gain observed in the HFD-AFA mice compared to HFD controls. The absence of effects on caloric intake led us to exclude the involvement of the product in the central mechanisms regulating food intake. 

It is known that the HFD-induced body weight increase is due to hyperplasia and hypertrophy of visceral adipose tissue [39,40]. Accordingly, our obese control mice showed a higher VAT weight, fat index and adipocytes mean area than the lean controls, confirming hypertrophy of adipose tissue. Interestingly, mice fed with a HFD supplemented with AFA showed a significant reduction in hypertrophy in comparison with obese animals, indicating that AFA extract is able to prevent the HFD-dependent fat-mass accumulation. 

VAT expansion is supported by lipogenesis involving both de novo synthesis and increased uptake of triglycerides and fatty acids [41]. Many transcription factors regulate lipogenesis including the peroxisome proliferator-activated receptor gamma (PPAR-γ) and the Sterol Regulatory Element-Binding Protein (SREBP)-1c. Moreover, lipogenic-related enzymes, such as fatty acid synthase (FAS), play important roles in lipid accumulation during lipogenesis. PPAR-γ is considered as one of the main activators of lipogenesis; in fact, PPAR-γ inhibitors were shown to be useful tools to treat obesity [42]. SREBP-1c is expressed in tissues in which fatty acid de novo synthesis is highly active, such as liver and adipose tissue. When activated, SREBP-1c translocate into the nucleus and act as transcriptional activators of genes mediating both cholesterol and lipid synthesis, including FAS [43]. Fatty acid transporter (FAT)-P is a membrane protein, mediating fatty acid uptake from extracellular milieu. It has been reported that the upregulation of FAT-P mRNA increases fatty acid uptake in the adipose tissue and livers of obese mice [44]. 

Our results showed that PPAR-γ, SREBP-1c, FAS and FAT-P genes were upregulated in the visceral adipose tissue of obese mice, confirming their involvement in the HFD-induced fat accumulation. For the first time our results revealed the epigenetic potential of microalgae extracts because the AFA diet was able to prevent the HFD-induced upregulation of SREBP-1c, FAS and FAT-P in the adipose tissue. The anti-obesogenic effects exerted by AFA were reflected by a significant reduction in both plasma triglyceride concentrations and fat liver accumulation in HFD-AFA mice compared to obese controls. The steatosis prevention could be related to the SREBP-1c downregulation observed in HFD-AFA livers. In fact, the SREBP-1c-dependent pattern upregulation is known as one of the main molecular mechanisms in nutrient-induced hepatic steatosis [45]. Various phytocompounds contained in AFA extract, such as omega-3 [46], AFA-phycocyanins [47] and chlorophyll [48], have been reported to have anti-adiposity potential by acting as positive modulator of lipid metabolizing gene expression. As such, we can speculate that these microalgae compounds are responsible for the observed hypolipidemic action.

C57BL/6J mice fed long-term HFD develop hyperglycaemia, hyperinsulinemia, impaired glucose, and insulin-resistance [49]. Here, we showed that AFA supplementation exerts positive impact also on the HFD-induced impaired glucose metabolism, as shown by the significantly reduced fasting glycaemia and insulinaemia and strongly improved glucose tolerance and insulin sensitivity in HFD-AFA mice compared to the obese controls. Moreover, the HOMA-IR, an index of IR, and the protein expression of insulin receptor in the liver resulted significantly reduced in obese mice fed AFA, confirming that microalgae extracts can effectively prevent the insulin resistance due to excess fat consumption. Consistent with our results, AFA extracts have been reported to be able to reduce blood glucose levels in Iranian type 2 diabetes patients [50] and to ameliorate basal glycaemia in streptozotocin-induced diabetic rats [51].

Chronic inflammation and oxidative stress are among the mechanisms responsible for abnormal glucose homeostasis due to a chronic consumption of a hyperlipidic diet. Indeed, HFD dependency increased the activation of TNF-α and NFkB involved in the mechanisms, leading to an altered insulin signalling [52,53]. Several compounds enriched in KlamExtra^®^ have high anti-inflammatory and anti-oxidant properties [14,54,55,56,57]. So, we evaluated the hypothesis that the AFA euglycemic effect is linked to the improvement of the inflammatory response and oxidative stress in the liver and adipose tissue of obese animals. The observation that AFA extract prevented the HFD-induced liver inflammation by strongly reducing gene expression of pro-inflammatory IL-6 and TNF-α cytokines, as well as the protein expression of NFkB, and by increasing the expression of the anti-inflammatory IL-10 protein, provides evidence for AFA extract’s ability to ameliorate glucose dysmetabolism by reducing inflammatory status. Consistent with our results, phycocyanobilin, one of the AFA phytocompounds [57], has been reported to exert anti-inflammatory effects by inhibiting the NFkB-dependent inflammatory signal pathways [58]. 

Oxidative stress is another mechanism associated with impaired insulin signalling [59], hepatic steatosis [60] and adiposity [61]. 

We observed that the AFA diet was able to strongly reduce ROS levels both in the liver and adipose tissue. On the other hand, NFkB is also involved in the control of ROS generation [62] and carotenoids, phycoerythrins, and phycocyanins, which are present in AFA composition, have strong antioxidant properties. Accordingly, microalgae extract strongly ameliorated the HFD-induced SOD activity impairment by bringing back its levels to that observed in lean mice. In line with our results, in vitro and in vivo studies have revealed the anti-inflammatory and antioxidant properties of Klamin^®^ and AphaMax^®^, the two AFA extracts included in KlamExtra^®^ formulation [14,20,63]. Therefore, it is possible to hypothesise that the AFA extract is also able to ameliorate insulin resistance, steatosis and adiposity by reducing oxidative stress. 

### 4.2. Does AFA Accelerate the Reversion of the Obesity-Related Dysmetabolism Obtained by Switching from HFD vs. STD?

Switching a high-fat diet for a normocaloric diet could reverse HFD-induced metabolic abnormalities [64,65], although the success of the diet seems to depend on the long-standing consumption of an HFD [9,66], or by the adherence to the diet which can be “optimised” by accelerating the achievement of the weight loss goals [5]. 

As such, another goal of our study was to evaluate the ability of AFA to potentiate the reversion of the HFD-induced dysmetabolism when the HFD diet was switched to a standard diet (STD). 

In our experimental conditions, AFA’s impact on several dysfunctions, including body weight loss, hepatic steatosis, and adiposity, could not be revealed because they were fully recovered by standard diet alone. However, further experiments are necessary to clarify these results, considering for example, a shorter experimental protocol.

Interestingly, AFA was able to potentiate the STD-induced reversion of glucose dysmetabolism as demonstrated by the significant reduction in basal glycaemia, HOMA-index and the improved insulin sensitivity observed in mice fed an STD supplemented with AFA in comparison to mice fed normal chow. Furthermore, we observed that AFA accelerates the recovery from inflammation and oxidative stress in liver and adipose tissue of STD-AFA mice confirming that AFA counteracts metabolic glucose impairment by acting as anti-inflammatory and antioxidant agent. 

## 5. Conclusions

In conclusion, by using an animal model of diet-induced obesity, we revealed that KlamExtra^®^ is able to prevent body weight gain, impaired glucose metabolism, hepatic steatosis and hypertrophy of the adipose tissue induced by a chronic hyperlipidic diet. We also highlighted the AFA’s ability to accelerate the reversion of the HFD-induced glucose dysmetabolism by consuming a normocaloric diet. These beneficial effects appear to be linked to the AFA’s potential to positively modulate the expression of factors linked to inflammation, oxidative stress, and lipid metabolism. Although human studies are necessary to confirm the effects of AFA extract, the present study offers a new potential and natural approach to the treatment of obesity-related dysfunctions. 

## Figures and Tables

**Figure 1 cells-12-02706-f001:**
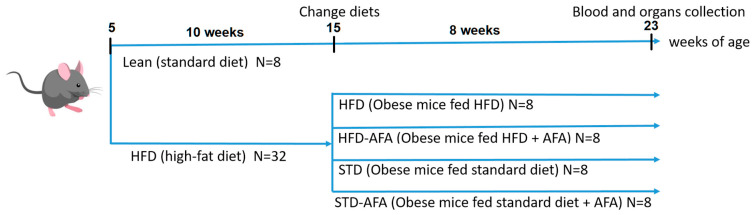
Graphical illustration of experimental design. C57BL/6J male mice were fed a standard diet for 18 weeks or a high-fat diet (HFD) for 10 weeks to establish obesity. Subsequently, HFD obese mice were divided in four sub-groups fed different diets for further 8 weeks: HFD and HFD-AFA groups fed, respectively, HFD alone or supplemented with AFA; STD and STD-AFA fed, respectively, standard diet alone or supplemented with AFA.

**Figure 2 cells-12-02706-f002:**
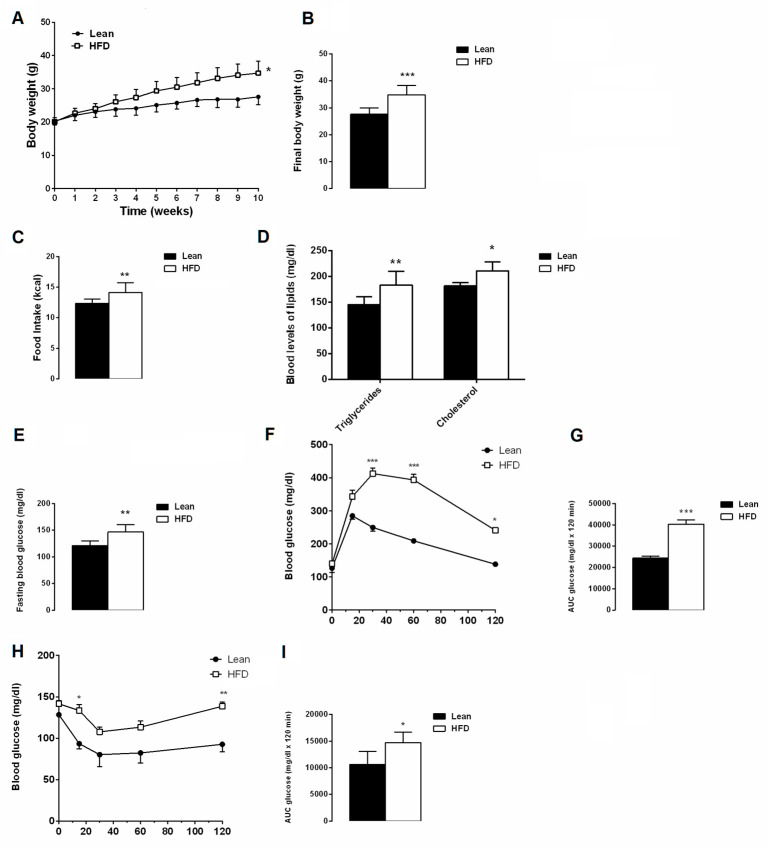
Effects of 10 weeks of HFD on metabolic parameters. (**A**) Body weight, (**B**) final body weight, (**C**) food intake expressed as energy intake (kcal/day), (**D**) blood levels of lipids, (**E**) fasting glycaemia, (**F**) glucose tolerance test (GTT), (**G**) area under the curve (AUC) for GTT, (**H**) insulin tolerance test (ITT), (**I**) area under the curve for ITT in lean and HFD mice groups. Data are the means ± S.E.M. (n Lean = 8; n HFD = 32). Asterisk denotes significant difference compared with the lean group (* *p* < 0.05; ** *p* < 0.01; *** *p* < 0.001).

**Figure 3 cells-12-02706-f003:**
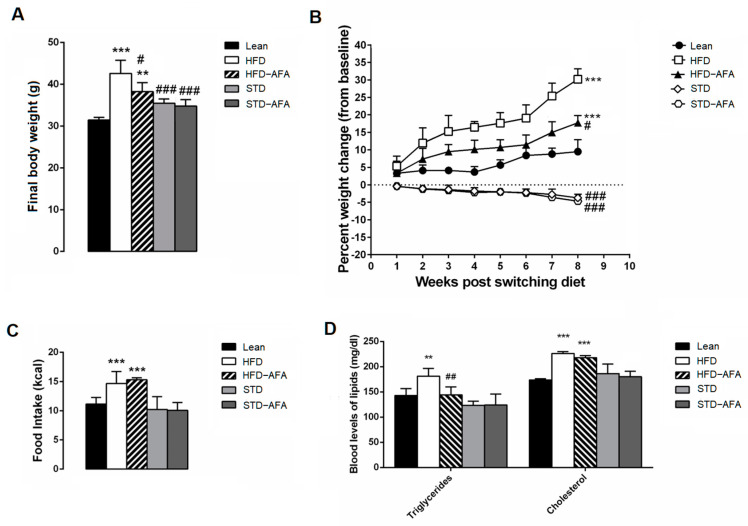
Effect of AFA on metabolic parameters. (**A**) Final body weight, (**B**) percentage of changes in body weight from switched diets, (**C**) energy intake (kcal per day), (**D**) blood lipid concentration in lean, HFD, HFD-AFA, STD and STD-AFA mice. Data are the means ± S.E.M. (n = 8/group). Asterisk denotes significant difference compared to the lean group (** *p* < 0.01; *** *p* < 0.001); hash denotes significant difference compared to the HFD group (# *p* < 0.05; ## *p* < 0.01; ### *p* < 0.001).

**Figure 4 cells-12-02706-f004:**
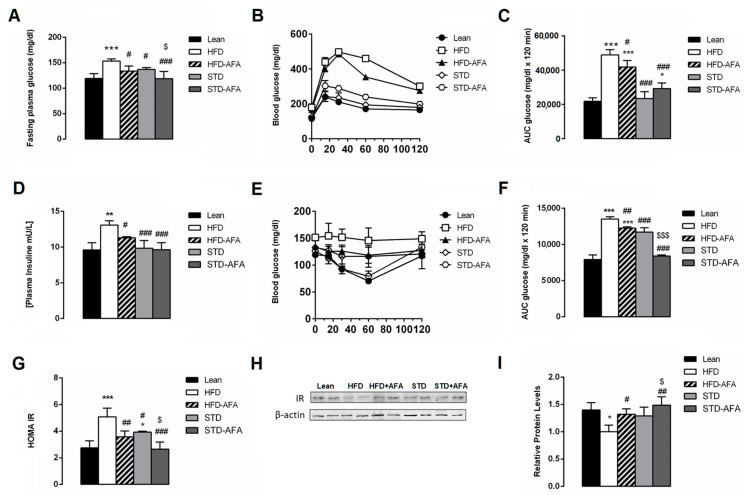
Effects of AFA on glucose metabolism. (**A**) Fasting glycaemia, (**B**) glucose tolerance test (GTT), (**C**) area under the curve (AUC) for GTT, (**D**) insulin tolerance test (ITT), (**E**) area under the curve for ITT, (**F**) plasma insulin levels, (**G**) HOMA index, (**H**) protein expression levels of insulin receptor and β-actin in liver and (**I**) densitometric quantification in Lean, HFD, HFD-AFA, STD and STD-AFA mice. Data are the means ± S.E.M. (n = 8/group). Asterisk denotes significant difference compared with the lean group (* *p* < 0.05; ** *p* < 0.01; *** *p* < 0.001); hash denotes significant difference compared with the HFD group (# *p* < 0.05; ## *p* <0.01; ### *p* < 0.001); dollar denotes significant difference compared with the STD group ($ *p* < 0.05; $$$ *p* < 0.001).

**Figure 5 cells-12-02706-f005:**
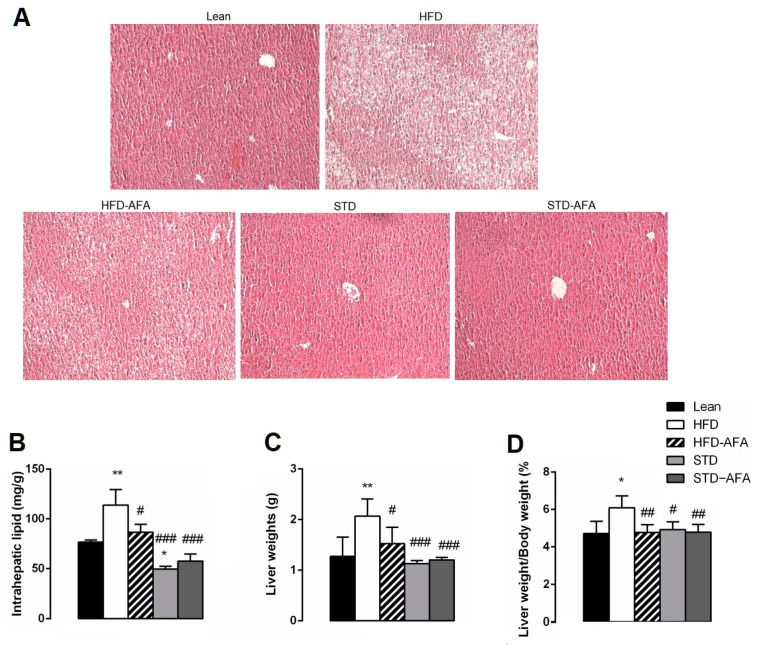
Effects of AFA on hepatic steatosis. (**A**) Histological cross-sections of the liver stained with haematoxylin and eosin (original magnification: ×100); (**B**) intrahepatic lipid content; (**C**) liver weight; (**D**) liver weight/body weight ratio in lean, HFD, HFD-AFA, STD and STD-AFA mice. Data are the means ± S.E.M. (n = 8/group). Asterisk denotes significant difference compared with the lean group (* *p* < 0.05; ** *p* < 0.01); hash denotes significant difference compared with the HFD group (# *p* < 0.05; ## *p* <0.01; ### *p* < 0.001).

**Figure 6 cells-12-02706-f006:**
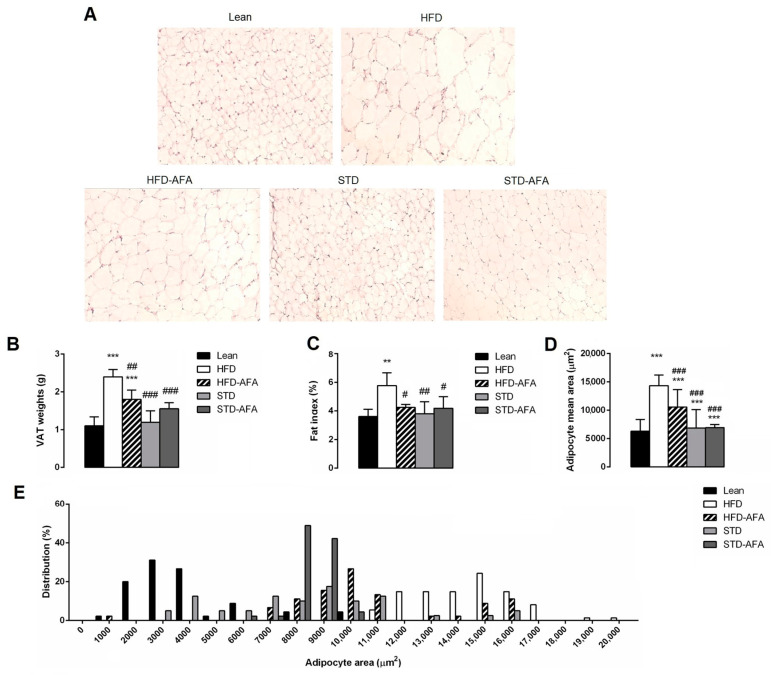
Effects of AFA on adiposity. (**A**) Histological cross-sections of adipose tissue stained with haematoxylin and eosin (original magnification: ×200); (**B**) VAT weight; (**C**) VAT weight normalised to body weight (fat index); (**D**) Adipocyte area (**E**) and size distribution (%) in lean, HFD, HFD-AFA, STD and STD-AFA mice. Data are the means ± S.E.M. (n = 8/group). Asterisk denotes significant difference compared with the lean group (** *p* < 0.01; *** *p* < 0.001); hash denotes significant difference compared with the HFD group (# *p* < 0.05; ## *p* <0.01; ### *p* < 0.001).

**Figure 7 cells-12-02706-f007:**
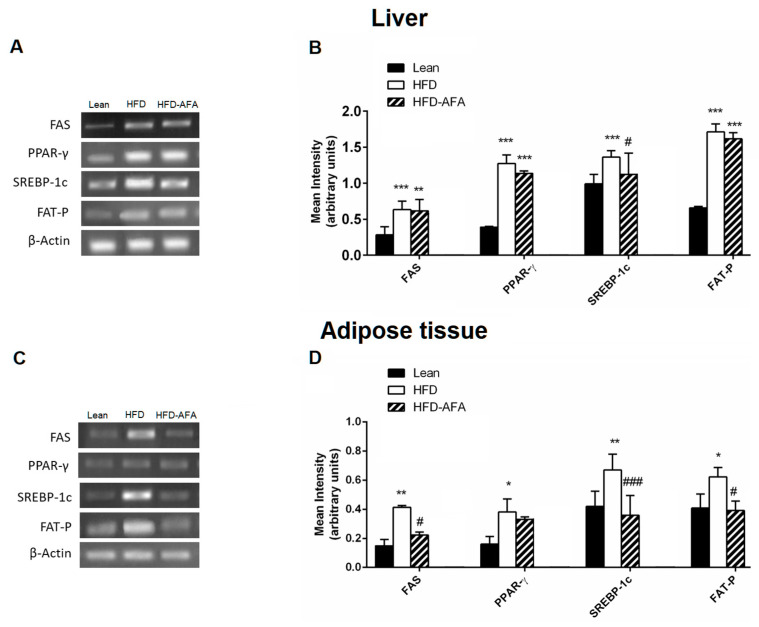
Effects of AFA on the expression of genes involved in lipid metabolism. Representative images of the RT-PCR results and mRNA levels of FAS, PPAR-g, SREBP-1c and FAT-P in the livers (**A**,**B**) and adipose tissues (**C**,**D**) of lean, HFD and HFD-AFA mice. Data are the means S.E.M. (n = 8/group). Asterisk denotes significant difference compared with the lean group (* *p* < 0.05; ** *p* < 0.01; *** *p* < 0.001); hash denotes significant compared with the HFD group (# *p* < 0.05; ### *p* < 0).

**Figure 8 cells-12-02706-f008:**
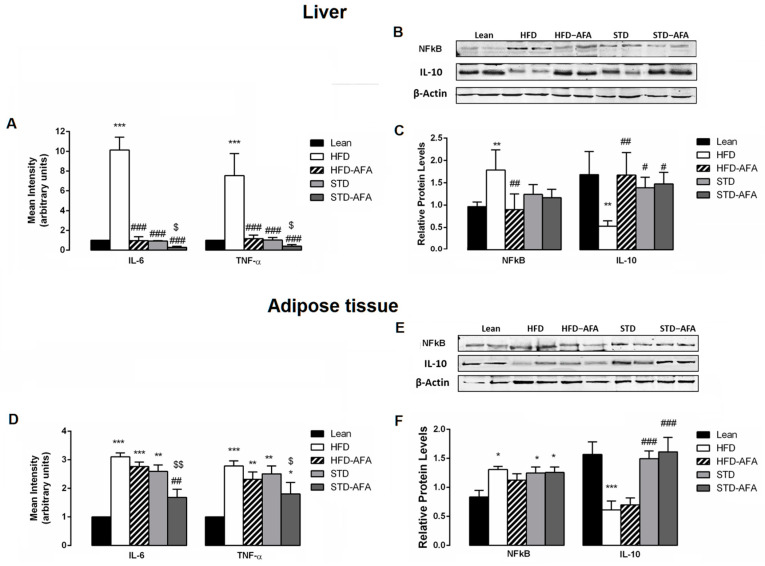
Effects of AFA on inflammation. IL-6 and TNF-α gene expression changes in livers (**A**) and adipose tissue (**D**) of lean, HFD, HFD-AFA, STD and STD-AFA mice. Representative Western blot bands of hepatic (**B**) and adipose tissue (**E**) NFkB, IL-10 and β-actin protein expression; densitometric analysis of hepatic (**C**) and adipose tissue (**F**) NFkB and IL-10 protein levels normalised for β-actin levels in Lean, HFD, HFD-AFA, STD and STD-AFA mice. Data are the means ± S.E.M. (n = 8/group). Asterisk denotes significant difference compared with the lean group (* *p* < 0.05; ** *p* < 0.01; *** *p* < 0.001); hash denotes significant difference compared with the HFD group (# *p* < 0.05; ## *p* <0.01; ### *p* < 0.001); dollar denotes significant difference compared with the STD group ($ *p* < 0.05; $$ *p* < 0.01).

**Figure 9 cells-12-02706-f009:**
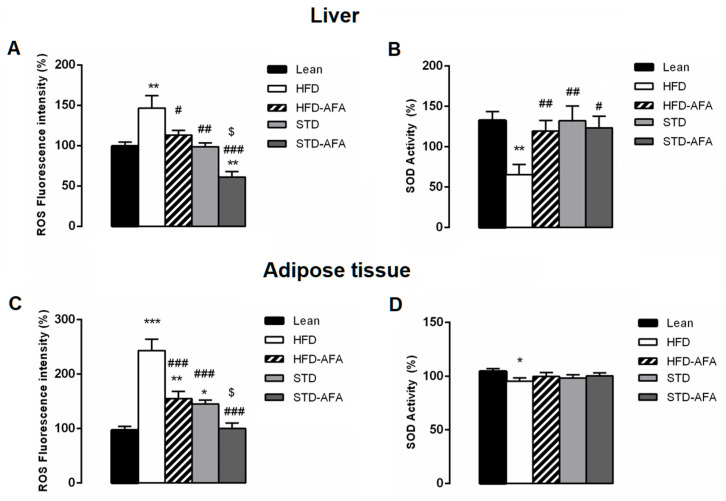
Effects of AFA on oxidative stress. Hepatic levels of ROS (**A**) and SOD activity (**B**) in lean, HFD, HFD-AFA, STD and STD-AFA mice. Adipose tissue levels of ROS (**C**) and SOD activity (**D**) in lean, HFD, HFD-AFA, STD and STD-AFA mice. Data are the means ± S.E.M. (n = 8/group). Asterisk denotes significant difference compared with the lean group (* *p* < 0.05; ** *p* < 0.01; *** *p* < 0.001); hash denotes significant difference compared with the HFD group (# *p* < 0.05; ## *p* <0.01; ### *p* < 0.001); dollar denotes significant difference compared with the STD group ($ *p* < 0.05).

**Table 1 cells-12-02706-t001:** Oligonucleotide sequence of primers for real-time PCR.

Gene	Forward Primer	Reverse Primer
TNF-α	5′-GCCCACGTCGTAGCAAACCAC-3′	5′-GGCTGGCACCACTAGTTGGTTGT-3′
IL-6	5′-TCCAGTTGCCTTCTTGGGAC-3′	5′-GTGTAATTAAGCCTCCGACTTG-3′
GAPDH	5′-GCCAAATTCAACGGCACAGT-3′	5′-AGATGGTGATGGGCTTCCC-3′

*TNF-α: Tumor Necrosis Factor-α; IL-6: Interleukin-6; GAPDH:Glyceraldehyde-3-phosphate dehydrogenase.*

**Table 2 cells-12-02706-t002:** Oligonucleotide sequence of primers for semi-quantitative RT-PCR.

Gene	Forward Primer	Reverse Primer
FAS	5′-TACTTTGTGGCCTTCTCCTCTGTAA-3′	5′-CTTCCACACCCATGAGCGAGTCCAGGCCGA-3′
PPAR-γ	5′-GGGCTGAGGAGAAGTCACAC-3′	5′-TCAGTGGTTCACCGCTTCTT-3′
SREBP-1c	5′-GGAGACATCGCAAACAAGC-3′	5′-GGTAGACAACAGCCGCATC-3′
FAT-P	5′-CGCCGATGTGCTCTATGACT-3′	5′-ACACAGTCATCCCAGAAGCG-3′

*FAS: fatty acid synthase, PPAR-γ: Peroxisome Proliferator-Activated Receptor-gamma; SREBP-1c: Sterol Regulatory Element-Binding Protein-1c; (FAT)-P: Fatty acid transporter-P.*

## Data Availability

The supporting data presented in this study are available from the corresponding author on reasonable request.

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
