# Peer review of "Positive Impacts of Aphanizomenon Flos Aquae Extract on Obesity-Related Dysmetabolism in Mice with Diet-Induced Obesity"

_cells, 2023, doi:10.3390/cells12232706_

Round 1
Reviewer 1 Report
Comments and Suggestions for Authors
In this paper, Terzo et al presented the positive impact of an AFA extract on the obesity-related dysmetabolism in male mice with diet-induced obesity. They concluded that AFA supplementation prevents high-fat diet induced dysmetabolism and accelerates standard-diet-dependent recovery of glucose dysmetabolism.
1. It would be important to provide data-set at 15 weeks before change of diet or treatment for a comprehensive interpretation of the results at 23 weeks.
2. What is the justification to use male only mice in the obesity modeling?
3. This study would benefit from testing an additional obesity mouse model for a more generalizable conclusion.
4. The quality of the liver images in Fig 4 is suboptimal.
5. In addition to the mRNA level, providing the relevant protein level of the investigated markers should be helpful.
6. In the conclusion, the authors state that "there beneficial effects appear linked to the epigenetic potential of AFA", where are the data in the manuscript to support this statement.
Comments on the Quality of English LanguageThe current version of the paper is readable. The quality of English language should improve with minor editing.
Author Response
We thank you for your suggestions and comments.
- It would be important to provide data-set at 15 weeks before change of diet or treatment for a comprehensive interpretation of the results at 23 weeks.
REPLY: Thank for your comment. In truth, at the end of 10 weeks of HFD, we have performed some in vivo analysis (body weight, food intake, basal glycaemia, GTT, ITT, lipidaemia) in Lean and HFD groups, to confirm the development of the HFD-related dysmetabolism. Anyway, we did not show them, because, usually these data are used to evaluate the changes of metabolic parameters in the same animal, before and after the treatment (intra-group analysis). In our study, we just wanted to analyse AFA effects among the groups of mice differently fed (inter-group analysis). Anyway, we accept your suggestion and we have added a new figure as supplementary data (Fig. S1) regarding our analysis at the end of 10 weeks of HFD (15 weeks of age). We have added in the text a sentence, referred to this analysis and to the new supplementary figure (Fig. 1S)
- What is the justification to use male only mice in the obesity modeling?
REPLY: Thank for your question. One of the most common reason for choosing only male mice in our studies is related to the hormone variability linked to the reproductive cycle of the female animals. Because of the hormonal differences, male and female individuals are not equally affected by HFD-induced metabolic alterations; in fatc, different studies (and our experience also) demonstrat that male C57BL/6 mice are more susceptible to hypercaloric diet, they gain more weight and develop higher glucose impairment, and dyslipidaemia than females.
- This study would benefit from testing an additional obesity mouse model for a more generalizable conclusion.
REPLY: I agree with you that to evaluate AFA beneficial effects in other additional obesity mouse model could add more information about its efficacy. Unfortunally, in this period we are not able to perform these analysis (because the costs and the long time to request Ministerial authorization for animal experimentation), but the future plains involve your suggestion.
- The quality of the liver images in Fig 4 is suboptimal.
REPLY: Thanks for your attention. We have tried to ameliorate the resolution of the liver images. Because some problems in the original captures of the images, we cannot operate better than this.
- In addition to the mRNA level, providing the relevant protein level of the investigated markers should be helpful.
REPLY: We agree with you, but in this specific case, our intention was to highlight AFA ability to induce changes in the expression of the genes linked to lipid metabolism and so to speculate on its potential epigenetic potential (as discussed in pag 14, lines 464-468 and 473-475).
- In the conclusion, the authors state that "there beneficial effects appear linked to the epigenetic potential of AFA", where are the data in the manuscript to support this statement.
REPLY: Thank you so much for this observation. The sentence has been changed in the conclusions.
Reviewer 2 Report
Comments and Suggestions for Authors
The authors evaluated the KlamExtra effects the changes on the glucose, lipid profile, inflammation, oxidative stress and hepatic and adipose tissue effects and abnormal lipid metabolizing gene expression induced by HFD.
The paper is interesting, materials & methods used are robust, however the authors could include more information regarding the extract used and relate the identified effects to its constituents.
Author Response
Thanks for the positive comments. Sure, to including more information regarding the KlamExtra composition would be helpful to comment and support the observed effects. Unfortunately, the composition of the extract is blinded by patent. The only information from the company regards the origin of KlamExtra, that is a combination of two patented AFA extracts: Klamin® and AphaMax®. The composition of AphaMax has been recently analyzed by Nuzzo et al. and we have reported in the manuscript the related references.
Round 2
Reviewer 1 Report
Comments and Suggestions for Authors
The authors addressed the reviewer's comments partially.
1. While it is helpful to provide data on 15 weeks (10 weeks after HFD initiation), data should be incorporated into the manuscript and discussed accordingly. Though the authors focused on inter-group comparison, longitudinal analysis is also important for proper interpretation of the findings.
2. The current set of liver histology images (Fig 4A) unfortunately are not readable.
Comments on the Quality of English LanguageCompared to the 1st version, English language has been improved to some degree.
Author Response
- While it is helpful to provide data on 15 weeks (10 weeks after HFD initiation), data should be incorporated into the manuscript and discussed accordingly.
Though the authors focused on inter-group comparison, longitudinal analysis is also important for proper interpretation of the findings.
REPLAY: We thanks the revisor for this further suggestion. We have added in the manuscript the figure about the analysis at the end of the 10 weeks of HFD (now, Figure 2) and the relative comments in the Results and Discussion.
- The current set of liver histology images (Fig 4A) unfortunately are not readable.
REPLY: We tried to perform new detections of the liver images, acquiring in higher definition. Moreover we have divided Fig.4 in two figures (now, figure 5 for livers and figure 6 for adipose tissues). We hope that in this way, with the liver images larger and more defined, they could be readable.

Round 3
Reviewer 1 Report
Comments and Suggestions for Authors
The authors addressed this reviewer's comments partially and the manuscript has thus been improved. No more questions raised.
Author Response
Dear Revisor,
Thank you for your support in improving our manuscript.